# Cancer-Specific Immune Prognostic Signature in Solid Tumors and Its Relation to Immune Checkpoint Therapies

**DOI:** 10.3390/cancers12092476

**Published:** 2020-09-01

**Authors:** Shaoli Das, Kevin Camphausen, Uma Shankavaram

**Affiliations:** Bioinformatics Core Facility, Radiation Oncology Branch, National Cancer Institute, National Institutes of Health, Bethesda, MD 20892, USA; shaoli.das@nih.gov (S.D.); camphauk@mail.nih.gov (K.C.)

**Keywords:** immune signature, immune checkpoint therapies, immune prognostic score

## Abstract

To elucidate the role of immune cell infiltration as a prognostic signature in solid tumors, we analyzed immune-function-related genes from four publicly available single-cell RNA-Seq data sets and twenty bulk tumor RNA-Seq data sets from The Cancer Genome Atlas (TCGA). Unsupervised clustering of pan-cancer transcriptomic signature showed two major immune function types: one related to NK-, T-, and B-cell functions and the other related to monocyte, macrophage, dendritic cell, and Toll-like receptor functions. Kaplan–Meier analysis showed differential prognosis of these two groups, dependent on the cancer type. Our analysis of TCGA solid tumors with an elastic net model identified 155 genes associated with disease-free survival in different tumor types with varied influence across different cancer types. With this gene set, we computed cancer-specific prognostic immune score models for individual cancer types that predicted disease-free and overall survival. Validation of our model on available published data of immune checkpoint blockade therapies on melanoma, kidney renal cell carcinoma, non-small cell lung cancer, gastric cancer and bladder cancer confirmed that cancer-specific higher immune scores are associated with response to immunotherapy. Our analysis provides a comprehensive map of cancer-specific immune-related prognostic gene sets that are associated with immunotherapy response.

## 1. Introduction

Immunotherapy recently has gained much attention as a viable treatment for multiple cancer types. Immune checkpoint blockade therapies targeting cytotoxic T-lymphocyte-associated antigen 4 (CTLA-4) or programmed cell death protein 1 (PD1) have shown efficacy in many cancers, like metastatic melanoma and non-small cell lung carcinoma [1,2,3,4]. These therapies try to stimulate a cytotoxic T-cell antitumor immune response by inhibiting PD1 or CTLA-4, which are negative regulators of T-cell activity [5,6]. However, despite immune checkpoint inhibitor therapies’ immense potential, not all patients’ tumors respond well to them [7]. This variable clinical response has been linked to patients’ genomic characteristics, like tumor mutation burden, microsatellite instability, and tumor microenvironment [8,9]. A particularly important aspect of the tumor microenvironment is the infiltration of various immune cell types, which is linked to patients’ prognosis [10]. A study from The Cancer Genome Atlas (TCGA) consortium explored the transcriptomic signature of the immune infiltration landscape in a pan-cancer context and classified solid tumors’ immune microenvironment into six types depending on enrichments in specific immune functions, like wound healing, interferon gamma response, inflammatory response, lymphocyte depletion, immunologically quiet, and transforming growth factor beta (TGF-beta) response phenotypes [9]. However, from this study, it is evident that tumor immune microenvironments have different compositions in different cancer types. In the present study, we set out to find the specific immune signatures that could be linked to patient prognosis in different cancer types. We used a pooled immune signature gene set curated from published literature. Analyzing single-cell RNA-Seq (scRNA-Seq) data from five solid tumor types, we found that our pooled immune signature gene set could characterize the tumor immune landscape. Our analysis on twenty TCGA tumor types demonstrated that good prognosis is associated with tumor-specific immune signature, with some overlap between tumors. Our analysis identified immune-function-associated genes that have prognostic significance specific to twenty different cancer types. We derived cancer-specific prognostic immune scores based on these. By analyzing published immune checkpoint inhibitor treatments in melanoma, bladder cancer, renal cell carcinoma, non-small cell lung cancer and gastric cancer, we showed that our cancer-specific immune score models could predict response to these immunotherapies.

## 2. Results

### 2.1. Analysis of Tumor scRNA-Seq Data Identifies Major Clusters with a Distinct Immune Signature

Tumor transcriptomic profiling at the single-cell level enables precise characterization of immune cell infiltrations. We analyzed scRNA-Seq data from five solid tumor types—melanoma [11], breast cancer [12], glioblastoma [13], head and neck cancer [14], and colorectal cancer [15]—to identify the tumors’ dominant immune functional signatures. First, we used unsupervised t-distributed stochastic neighbor embedding (t-SNE) clustering to identify distinct cell populations (Figure 1a, Appendix A) while simultaneously performing single sample gene set enrichment analysis (ssGSEA) at the single-cell level with our curated immune signature gene sets combined from four sources (Leukocyte signature Matrix 22 or LM22 [16], Leukocyte signature Matrix 7 or LM7 [17], *^1 1^* Immune cell gene Signature or ImSig [18], and the NanoString immune signature panel (https://www.nanostring.com)). The major clusters found from unsupervised clustering of ssGSEA scores of the single cells were assigned to the immune functions that were highly and specifically enriched in the respective clusters (Figure 1b, Appendix A). We calculated the fraction of cells assigned to a specific immune function (e.g., NK and T cells, identified from ssGSEA) in the cell population clusters identified from the unsupervised t-SNE analysis (Figure 1c, Appendix A). A cell population (from t-SNE) was assigned to an immune function if more than 70% of its cells had high enrichment of that immune function. For melanoma and breast cancer, two major immune function clusters were associated with enrichment of NK and T cells and B cells and with enrichment of macrophages, monocytes, dendritic cells (MoMaDC), eosinophils, neutrophils, chemokines, and cell cycle functions, respectively (Figure 1b, Appendix A). Within the first cluster (NK and T cells and B cells), enrichment of B-cell functions formed a distinct cluster (annotated in purple in Figure 1b,c and Appendix A). For glioblastoma and head and neck cancers, we identified only two major immune function clusters, one associated with NK and T and B cells and the other associated with mixed cell function (MoMaDC/neutrophils/cell cycle functions) (Appendix A). For these cancer types, enrichment of B-cell functions did not form any distinguished signature, unlike in melanomas and breast cancers. For colorectal cancer, NK and T cell enrichments formed a distinct cluster, while the MoMaDC and B cell enrichments were part of the other major cluster, along with enrichments of some non-specific functions (chemokines, adhesion, platelets etc.) (Appendix A). This second major cluster could be subdivided into three distinct clusters: one with enrichment of MoMaDC, another one with enrichment of B cells, and the third one with non-specific functions. Figure 1 shows the analysis of immune signatures from scRNA-Seq data for the melanoma data set [11]. Our analysis identified cell population clusters associated with NK- and T-cell functions (cell clusters 0, 1, 5, 6, and 12), B-cell functions (cell clusters 2 and 14), and MoMaDC/neutrophils/cell cycle functions (cell clusters 7, 9, and 10). We checked our annotation’s validity by comparing it against the cell type annotation from the original paper by Tirosh et al. [11]. Our immune function annotation of NK/T-, and B-cell populations matched with the original paper with high specificity and sensitivity (94% and 96% specificity for NK/T and B cells, respectively; 98% and 99% sensitivity for NK/T and B cells, respectively). One of the cell populations with high enrichment of MoMaDC/neutrophils/cell cycle functions (cell cluster 7) matched the macrophage cell annotation from the original paper with 98% specificity and 99% sensitivity, while 99% of the other two cell population with similar enrichment (cell clusters 9 and 10) were annotated as malignant in the original paper. Similar analysis and cell type annotation were done for breast cancer, glioblastoma, head and neck, and colorectal cancer scRNA-Seq data sets (Appendix A). Overall, the results from the ssGSEA analysis on tumor scRNA-Seq data sets from multiple different cancer types using our curated immune gene sets indicates towards two to three major immune functional enrichments within the tumor microenvironment that were associated with NK- and T-cells, B cells, and MoMaDC functions, respectively.

### 2.2. Pan-Cancer Analysis of Bulk Tumor RNA-Seq Data Identifies Two Major Immune Clusters Associated with Prognosis 

As the ssGSEA analysis of tumor scRNA-Seq data using our curated immune gene set could identify distinct immune cell type enrichments within tumor microenvironment, next, we performed the same analysis on the bulk RNA-Seq data of 20 solid tumor types from TCGA. Unsupervised clustering of 8457 tumor samples was performed for the 23 immune gene sets with high variance in the ssGSEA scores across tumor samples (variance greater than mean). Two major immune function clusters were identified: one with high enrichment of NK-cell, T-cell, and B-cell functions (which we termed as immune cluster 1), and the other with high enrichment in MoMaDC, microglia, and Toll-like receptor (TLR) functions (immune cluster 2, Figure 2a). This result agrees with our previous analysis of scRNA-Seq data sets, which also identified distinct immune cell populations associated with NK/T/B cells and MoMaDC functions. To see if these two distinct immune function clusters are associated with any differences in patient prognosis for each of the 20 cancer types, we performed Kaplan–Meier analysis of disease-free and overall survival for TCGA patient samples stratified into immune cluster 1 (NK-, T-, and B-cell enrichment) and immune cluster 2 (MoMaDC, TLR, and microglial enrichment). Notably, for some tumor types (GBM, LGG, OV, BLCA, PAAD, and BRCA), immune cluster 1 showed better disease-free or overall survival (*p* < 0.1, Figure 2b). Conversely, for some other tumor types (KIRC, KIRP, UCEC, UVM, and SKCM), immune cluster 2 showed better disease-free or overall survival (*p* < 0.1, Figure 2c). This result shows that prognostic immune signatures vary between different tumor types.

### 2.3. Analysis of Cancer-Specific Immune Signature Genes 

As seen from previous analysis, prediction of better prognosis varies among cancer types and depends on the enrichment of either cluster 1 or 2 immune functions. This observation prompted us to derive cancer-specific prognostic immune signatures instead of a pan-cancer immune signature. We were interested in finding a minimal set of genes associated with immune functions that can still predict prognosis for individual cancer types. Our curated immune-function-associated gene set comprised 1351 genes. We reasoned that all the genes comprising the immune function gene sets would not be equally important in predicting patient prognosis. We used elastic-net-model-based gene selection for predicting disease-free survival (as described in the Methods) to find the minimal gene subsets for prognostic markers in each cancer type. The selected subsets for each of the 20 cancer types are shown in Appendix A. To further reduce the number of prognostic genes in each cancer type, we performed individual gene-wise Kaplan–Meier analysis on the shortlisted genes to select only those that were significantly associated with disease-free survival in that cancer type. By selecting immune signature genes for all 20 types, we obtained a set of 155 relevant genes associated with prognosis in different tumor types. Figure 3a shows the map of all 155 genes and their associations with prognosis in the 20 TCGA tumor types. Some genes that are associated with worse prognosis in one tumor type are associated with good prognosis in others; e.g., EIF3H is associated with worse prognosis in LIHC and KIRP but better survival in LGG. Notably, different tumor types share poor prognosis-associated genes more often, while the genes associated with better prognosis are rarely shared between tumor types. To check the expression pattern of our minimal prognostic immune signature genes (the set of 155 immune function associated genes) across all tumor types, we performed hierarchical clustering of these genes in the TCGA 20-cancer cohort. In the pan-cancer context across the 20 types, the expression of these 155 genes form two distinct clusters, as seen from the heat map and unsupervised clustering in Figure 3b. Immune signature genes that are associated with only poor prognosis in different tumor types (the gene cluster annotated with orange row side color) form a very distinct expression signature, as they are overexpressed in sample cluster red (red bar representing column annotations at the top of the heatmap), while the genes associated with either only good prognosis or with varied effect in different cancers are mostly overexpressed in sample cluster cyan (cyan bar representing column annotations at the top of the heatmap). Kaplan–Meier analysis of samples between these two clusters showed that in most cancer types (LGG, SKCM, LUAD, KIRC, KIRP, KICH, PRAD, PAAD, LIHC, THCA, BLCA), cluster red is significantly associated with poor disease-free survival (*p* < 0.05, Appendix A). In other tumor types (OV, BRCA, CESC, UCEC, UVM, HNSC, COADREAD, STAD, and GBM) we observed no significant differences in disease-free survival between the two clusters. This result shows that a poor prognosis signature is common between tumor types to some extent. Pathway enrichment revealed that the genes associated with poor prognosis in more than one cancer type are enriched for cell cycle functions (Figure 3c). Conversely, genes associated with only good prognosis were found to be enriched for interferon signaling and complement. These enriched pathways were common among several cancer types, including SKCM, LIHC, KIRP, HNSC, BRCA, and PRAD (Figure 3d).

### 2.4. Cancer-Specific Prognostic Immune Score Models Predict Disease-free and Overall Survival for 20 Cancer Types

To get a combined immune score, we calculated the sum of expression of the final selected genes (elastic net selection followed by Kaplan–Meier analysis), weighted by their respective elastic net coefficients. Figure 4a shows the schematic diagram of our computational pipeline. To demonstrate the utility of our model, Figure 4b–f shows the cancer-specific immune score signatures in GBM. With the elastic net selection, we found 31 immune-function-associated genes that predicted poor (less than six months) or better (≥12 months) disease-free survival. The better and poor disease-free survival were stratified based on the median disease-free survival of GBM patients from TCGA. The elastic-net-based gene selection was followed by individual gene-wise Kaplan–Meier analysis for each of the 31 genes to further select only the genes that are significantly associated with GBM patients’ disease-free survival. At this step, we further reduced the gene list to only 13 genes; Figure 4b shows the expression heat map for these 13 genes in the TCGA GBM cohort. The expression of these genes, weighted by their respective elastic net coefficient, is then combined into a unified immune score for GBM (as described in the Methods). From the TCGA data, we confirmed that this immune score can stratify GBM patients into better or worse disease-free and overall survival groups (Figure 4c,d, Kaplan–Meier *p* < 0.001, patients stratified by immune score greater than or less than median). High immune score predicted better disease-free and overall survival in patients, while low immune score predicted worse survival. Moreover, using multivariate Cox regression analysis on the TCGA GBM patients, we found that this immune score is an independent prognostic factor for disease-free survival along with two other known prognostic factors: patient age and MGMT methylation status (Figure 4e). To test our model in an independent patient cohort, we calculated the immune score using the same formula for the 13 gene signatures in the Repository of Molecular Brain Neoplasia Data (REMBRANDT) cohort for GBM patients [19]. Kaplan–Meier analysis of overall survival on the REMBRANDT patient cohort also showed that patients could be significantly stratified into better or worse survival groups by the immune score (Figure 4f, *p* < 0.05, immune score greater than or less than median). The same analysis done on all of the 20 TCGA cancer types showed that our cancer-specific immune score could significantly stratify patients into better/worse prognosis groups for disease-free and overall survival (Appendix A).

### 2.5. Cancer-Specific Immune Score Models Predict Response to Immune Checkpoint Inhibitor Treatment

As our cancer-specific immune scores could successfully predict patient survival in different cancer types, we sought to check whether they could also predict response to immunotherapy. The response to immune checkpoint inhibitor treatment is available in five cancer types—anti-PD1-treated melanoma [20,21,22,23,24], anti-PDL1-treated bladder urothelial cancer and renal cell cancer (ImVigor clinical trial [25], anti-PD1-treated renal cell cancer [26], anti-PD1 treated metastatic gastric cancer [27], and anti-PD1 treated non-small cell lung cancer [28]—from ten published data sets of cancer patients. We tried to predict objective response using our cancer-specific immune score model. In all these studies, objective response to these checkpoint inhibitors is documented according to the Response Evaluation Criteria in Solid Tumors (RECIST) guideline [29,30]. For the anti-PD1-treated melanoma [20,21,22,23,24], the immune score specific to SKCM was calculated using the patient transcriptomic data from these studies. Similarly, for the anti-PD1/PDL1-treated renal cell cancer patients [25,26], the immune score specific to KIRC was calculated; for the anti-PDL1-treated bladder urothelial tumor (bladder or ureter as tissue of origin) patients [25], the immune score specific to BLCA was calculated; for anti-PD1 treated non-small cell lung cancer patients, the immune score specific to LUAD was calculated; and for anti-PD1 treated gastric cancer patients, the immune score specific to STAD was calculated. Patients with complete response (CR) or partial response (PR) from these studies had a higher cancer-specific immune score compared to patients with stable disease (SD) or progressive disease (PD) that did not respond (Figure 5a–e). We could predict objective response (CR or PR) with the cancer-specific immune scores with an area under the curve (AUC) of 0.76, 0.67, 0.81, 0.61, and 0.59 for the melanomas (Hugo, Prat, Gide, Abril-Rodriguez, and Riaz data sets, respectively), 0.66 and 0.62 for the renal cell cancer (Miao and Mariathasan data sets, respectively), 0.69 for the bladder cancer (Mariathasan data set), 0.69 for gastric cancer (Kim data set), and 0.68 for the non-small cell lung cancer (Hwang data set) (Figure 5f). To investigate whether the cancer-specific immune score models (for melanoma, kidney, bladder, gastric, and non-small cell lung cancers) are important for predicting objective response in the respective cancers, we checked whether we could achieve similar prediction accuracy using an immune score model specific to another cancer type (a non-cancer-specific immune score). We tried to predict objective response in the anti-PDL1-treated bladder cancer from the ImVigor clinical trial data set using SKCM-specific and KIRC-specific immune scores. Notably, these non-cancer-specific immune scores did not show significant changes between patients whose tumors responded (CR or PR) and those whose tumors did not respond (SD or PD) (Appendix A). These non-cancer-specific immune score models performed no better in predicting immunotherapy response in bladder cancer patients than random predictions with AUC close to 0.5 did (Appendix A).

## 3. Discussion

The success of immune checkpoint inhibitor therapies in some metastatic tumors has opened a promising new avenue in cancer therapeutics. At the same time, it has become important to identify and prioritize patients who could benefit from such therapies. The composition of individual patients’ immune microenvironments is a critical determinant of prognosis and response to therapies. There has been comprehensive pan-cancer study on the tumor immune transcriptomic signature from the TCGA consortium [9]. Our analysis takes a different approach in this regard by identifying the cancer-specific prognostic immune signatures for 20 TCGA cancer types, comprising a minimal set of genes for each cancer type. Notably, our analysis demonstrates that each of the different cancer types has a unique immune signature of prognostic significance. We formulated cancer-specific prognostic immune scores that were significantly associated with better disease-free and overall survival in all cancer types and predicted response to immunotherapy.

Our analysis used a pooled immune signature of gene sets associated with different immune functions curated from published literature. scRNA-Seq enables characterization of immune infiltration in solid tumors at the single-cell level. Analysis of scRNA-Seq data from five solid tumor types showed that our pooled immune signature captures the immune infiltration signatures in these tumors. We applied the same immune signature gene sets to bulk tumor RNA-Seq data of 20 cancer types from TCGA and identified two major tumor clusters associated with enrichments of two distinct sets of immune functions: NK/T/B cells and MoMaDC/TLR response. Previously, Thorsson and colleagues identified an inflammatory tumor transcriptomic signature that had a high T helper 1 to T helper 2 ratio and was associated with a favorable outcome in TCGA pan-cancer cohort [9]. However, when we analyzed the prognostic immunologic signatures in a cancer-specific manner, we observed differences in immune signatures showing good prognosis in different tumor types. The prognostic immune signatures of some tumor types, like glioblastomas and gliomas, breast cancers, ovarian cancers, bladder cancers, and pancreatic cancers, were found to be close to each other but different from that of some other tumor types, like kidney cancers, uterine cancers, and melanomas. Notably, gliomas and pancreatic, ovarian, and breast cancers traditionally have been considered immunologically “cold” or immune-suppressed tumors, while melanomas and kidney cancers have been considered immunologically “hot” or T-cell-inflamed tumor types [31]. Our results imply that immunologically hot and cold tumor types have different pathways associated with better prognosis, indicating that they require different therapeutic approaches [32].

Our observation that different tumor types show different prognosis associated with immune signatures prompted us to identify a cancer-specific set of immune function genes of prognostic significance. Using elastic net and gene-wise Kaplan–Meier analysis, we identified a minimal set of genes for each cancer type. Our immune scores calculated using the cancer-specific prognostic gene sets could predict better disease-free and overall survival for each of the cancer types. Notably, when we compared the genes associated with good or poor prognosis from our analysis across 20 cancer types, we found that the genes associated with poor prognosis in cancers were more common between the cancer types. Common genes associated with poor prognosis in different cancer types were enriched with cell cycle functions, a common oncogenic process associated with proliferation. Interestingly, in a pan-cancer context, the genes mostly associated with poor prognosis were found to form a distinct cluster that could stratify patients into a poor survival group in most of the cancer types. The genes associated with good prognosis were not often shared between the cancer types; however, pathway enrichment of the genes associated with only good prognosis in cancers showed that enrichment of interferon response was commonly associated with good prognosis in multiple cancer types. This observation agrees with previous findings about type 1 interferon responses’ antitumor activity [33].

Our proposed cancer-specific immune score models were associated with response to immune checkpoint inhibitor treatments in different cancer types. Using published and clinical trial data sets, we confirmed in melanoma, bladder urothelial carcinoma, renal cell carcinoma, non-small cell lung cancer and gastric cancer that the higher cancer-specific immune scores were associated with complete or partial response to the treatments. Our cancer-specific immune score models could predict objective response to immunotherapies to some degree, while non-specific immune scores (for a different cancer type) could not predict the objective response better than a random prediction. This result also underscores the significance of cancer-specific immune signatures.

## 4. Materials and Methods 

### 4.1. Data Sources and Pre-Processing

We used publicly available scRNA-Seq data from patient tumor samples for five tumor types (melanoma, breast carcinoma, glioblastoma, head and neck cancer and colorectal cancer) [11,12,13,14,15]. Bulk RNA-Seq data for 20 tumor types—skin cutaneous carcinoma (SKCM), breast invasive carcinoma (BRCA), glioblastoma multiforme (GBM), lower-grade glioma (LGG), head and neck squamous cell cancer (HNSC), lung adenocarcinoma (LUAD), pancreatic adenocarcinoma (PAAD), prostate adenocarcinoma (PRAD), ovarian cancer (OV), cervical squamous cell carcinoma (CESC), bladder carcinoma (BLCA), uterine carcinoma (UCEC), renal cell carcinoma (KIRC), kidney papillary cell carcinoma (KIRP), kidney chromophobe (KICH), liver hepatocellular carcinoma (LIHC), thyroid carcinoma (THCA), stomach adenocarcinoma (STAD), colorectal carcinoma (COADREAD), and uveal melanoma (UVM)—were collected from TCGA [34].

The scRNA-Seq data for 19 melanoma patient samples were collected from Tirosh and colleagues’ published transcriptomic data of 4645 single cells [11] deposited in the Gene Expression Omnibus (accession: GSE72056). The scRNA-Seq data for 11 breast cancer patient samples were collected from Chung and colleagues’ published transcriptomic data of 549 single cells [12] deposited in the Gene Expression Omnibus (accession: GSE75688). The scRNA-Seq data for four glioblastoma patient samples were collected from Tirosh and colleagues’ published transcriptomic data of 3589 single cells [13] deposited in the Gene Expression Omnibus (accession: GSE84465). The scRNA-Seq data for 18 head and neck cancer patient samples were collected from Tirosh and colleagues’ published transcriptomic data of 5901 single cells [14] deposited in the Gene Expression Omnibus (accession: GSE103322). The scRNA-Seq data for 11 colorectal cancer patients were collected from Li and colleagues’ published transcriptomic data of 375 single cells [15] deposited in the Gene Expression Omnibus (accession: GSE81861) The scRNA-Seq data sets were processed with the R package Seurat. Log-normalized transcript per million was used to perform clustering of cell types via t-Distributed Stochastic Neighbor Embedding (t-SNE).

The bulk RNA-Seq data for 20 tumor types were downloaded as RSEM-processed RNA-Seq v2 files from cBioPortal (Firehose Legacy data sets) [35]. Corresponding patient clinical information containing follow-up data for overall and disease-free survival information was also collected from cBioPortal.

The patient data for melanoma, bladder urothelial carcinoma, renal cell carcinoma, gastric cancer, and non-small cell lung carcinoma tumors treated with anti-PD1/PDL1 were collected from published literature. Patient transcriptomic and clinical data with response to anti-PD1 treatments in melanoma were collected from five published studies [20,21,22,23,24]. Patient transcriptomic and clinical data for anti-PD1/PDL1 treatments in renal cancer were collected from Miao and colleagues’ and Mariathasan and colleagues’ studies, respectively [25,26]. Patient transcriptomic and clinical data for bladder urothelial carcinoma treated with anti–programmed death ligand 1 (PDL1) were collected from the IMvigor clinical trial data [25]. Patient transcriptomic and clinical data for metastatic gastric cancer treated with anti-PD1 was collected from the study by Kim and colleagues [27]. Patient transcriptomic and clinical data for anti-PD1 treated non-small cell lung cancer was collected from the study by Hwang and colleagues [28]. All transcriptomic data are normalized to log counts per million using EdgeR package.

### 4.2. Computational Analysis and Metadata Processing

We curated the signatures specific to immune functions from four published resources: the LM22 immune infiltration signature used in CIBERSORT [16], the LM7 immune infiltration signature derived by Tosolini and colleagues [17], the ImSig signature of solid tumor immune infiltration derived by Nirmal and colleagues [18], and the NanoString immune signature panel (https://www.nanostring.com). The combined 61 gene sets from these resources were used for single-sample Gene Set Enrichment Analysis (ssGSEA) [36], conducted using the R package Gene Set Variation Analysis, on the scRNA-Seq and bulk RNA-Seq tumor data sets. Unsupervised clustering of ssGSEA scores across tumors (for bulk RNA-Seq) or single cells (for scRNA-Seq) was performed using Spearman’s correlation coefficient as a similarity metric. Kaplan–Meier analysis was used to assess univariate survival, and a Cox proportional hazard model was used to assess multivariate survival. A two-sided Wilcoxon test was used to calculate *p*-values of differences in immune scores between treatment response groups in the immunotherapy data sets.

### 4.3. Calculation of Cancer-Specific Immune Scores 

The cancer-specific immune scores were calculated for 20 TCGA tumor types using bulk RNA-Seq transcriptomic data. As a first step, single-sample enrichments of 61 curated immune gene sets were calculated for each cancer type. Kaplan–Meier analysis was then performed for disease-free survival in samples from each cancer type, which were stratified into high or low enrichment (greater or less than median) of each of the 61 immune functions. The genes comprising the immune gene sets with significantly better or worse disease-free survival were used to find the minimal set of genes that could still predict disease-free survival in that cancer type. Next, an elastic net model was used on this set of genes, with the goal of predicting the short-term surviving (<12 months) versus long-term surviving patients (>24 months) using a reduced number of genes. Seventy percent of samples from the cancer type were used to train the elastic net model, while the remaining 30% were used for testing. Tenfold cross-validation was used to optimize the hyperparameters. The genes with non-zero coefficients from the elastic net model were selected as the shortlisted genes for the immune signature specific to the cancer type. To minimize the shortlisted genes further, gene-wise Kaplan–Meier analysis was performed to assess the association of disease-free survival with each gene and stratify the patient samples from that cancer type into high-expression (greater than median) or low-expression (less than median) groups. The genes showing a significant association with good or poor disease-free survival (*p* < 0.05) were chosen for the final immune score model for that cancer type. Finally, the cancer-specific immune score was calculated by the weighted sum of expression of the genes identified in the last step, with the corresponding elastic net coefficients for the genes used as weights.

### 4.4. Availability of Codes and Metadata

The codes and metadata used for the analysis presented in this paper are available from the GitHub repository: https://github.com/shaoli86/cancer-specific_immune_signatures.

## 5. Conclusions

In conclusion, our study indicates that immune signatures specific to different cancer types are significant for patient prognosis. It also serves as a framework for calculating cancer-specific immune scores for 20 cancer types that could be associated with prognosis and response to immunotherapies.

## Figures and Tables

**Figure 1 cancers-12-02476-f001:**
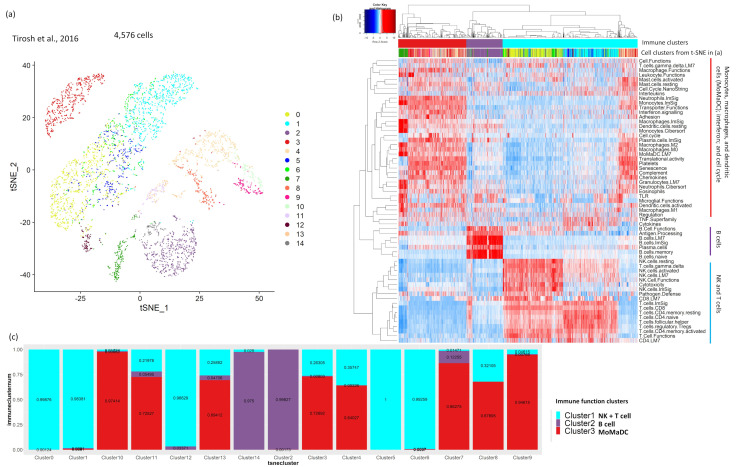
ssGSEA of scRNA-Seq data of melanoma tumor samples identifies three dominant immune-function-associated cell clusters. One of these clusters is enriched for monocytes, macrophages, dendritic cells, mast cells, and neutrophils; one is enriched for B-cell functions; and one is enriched for NK- and T-cell functions. (**a**) Analysis of a melanoma scRNA-Seq data set identifies 14 cell clusters from unsupervised t-SNE clustering. (**b**) These clusters were assigned to three distinct sets of immune functions using ssGSEA, as seen from the heat map (one related to monocytes, macrophages, dendritic cells, and Toll-like receptor (TLR); one related to B-cell functions; and one related to NK- and T-cell functions). (**c**) The stacked bar plot shows that cell clusters 0, 1, 5, 6, and 12 are enriched for NK- and T-cell functions; cell clusters 2 and 14 are enriched for B-cell functions; and cell clusters 7, 9, and 10 are enriched for functions related to monocytes, macrophages, and dendritic cells.

**Figure 2 cancers-12-02476-f002:**
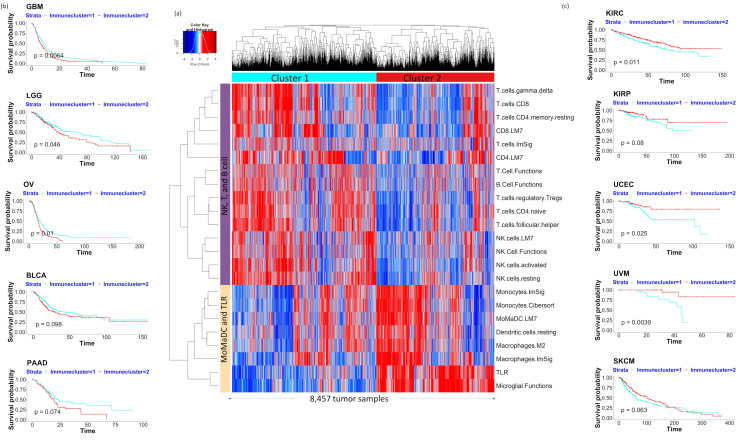
ssGSEA of bulk tumor RNA-Seq data from 20 TCGA tumor types. (**a**) Unsupervised clustering of ssGSEA scores of immune signatures across 20 cancer types identifies two distinct clusters: one enriched for NK-, T-, and B-cell functions, and the other enriched for macrophage, monocyte, dendritic cell, and TLR functions. (**b**) Cluster 1, which is enriched for NK-, T-, and B-cell functions, was associated with better survival (disease-free survival [DFS] or overall survival [OS]) for GBM, LGG, PAAD, BLCA, OV, and BRCA. (**c**) Cluster 2, which is enriched for TLR, microglial, macrophage, monocyte, and dendritic cell functions, was associated with better survival (DFS or OS) for UCEC, UVM, KIRC, KIRP, and SKCM.

**Figure 3 cancers-12-02476-f003:**
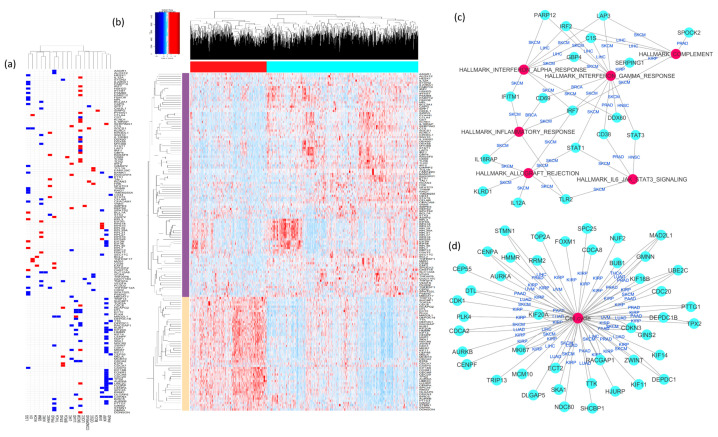
(**a**) Using a computational pipeline of an elastic net model and univariate Kaplan–Meier analysis for 20 TCGA tumor types, we identified 155 immune-signature-associated genes with prognostic significance in different cancers. The map shows these genes, color-coded as blue (negatively associated with prognosis (negative elastic net coefficient)) or red (positively associated with prognosis (positive elastic net coefficient)). (**b**) The heat map shows unsupervised clustering of expression of these 155 genes in tumor samples from 20 cancer types. The genes formed two major clusters (annotated with red and cyan colors on the columns). The samples in the red cluster have high expression of genes annotated with the beige side row color. Most of these genes are associated with poor prognosis in cancers, as seen from the map in (**a**). The samples from the cyan clusters have low expression of the poor prognosis genes annotated with the beige side row color but high expression of the genes annotated with the purple side row color. These genes had mixed prognosis. Some of them were associated with good prognosis, and some of them were associated with good prognosis in one cancer but poor prognosis in another cancer. (**c**,**d**) Pathway enrichment for (**c**) genes associated with better prognosis and (**d**) genes associated with worse prognosis in a pan-cancer context. We consider a gene to be associated with worse prognosis if it is associated with worse prognosis in more cancer types than cancer types where it is associated with better prognosis. For example, *MKI67* is associated with better prognosis in SKCM but worse prognosis in two other cancer types (KIRP and LIHC), so we consider it to be associated with worse prognosis in a pan-cancer context.

**Figure 4 cancers-12-02476-f004:**
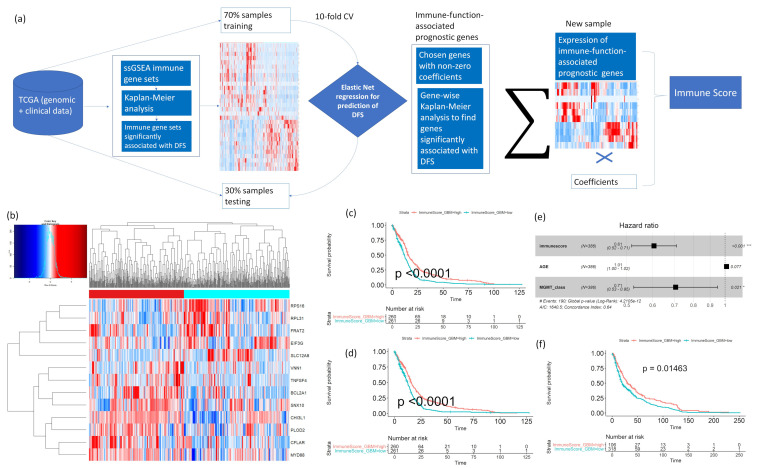
(**a**) The workflow for identifying immune-function-related prognostic genes and calculating a combined prognostic immune score for each TCGA cancer type is shown. For a given cancer type, ssGSEA is performed on the 61 curated immune gene sets, and Kaplan–Meier analysis is performed to reveal sets that significantly stratify patients into better or worse DFS based on the median ssGSEA score. To determine the minimum set of genes with prognostic significance, the genes associated with these filtered immune gene sets are pulled out and undergo expression modeling by an elastic net, enabling us to shortlist genes positively or negatively associated with DFS length. The elastic net model is trained on 70% of the samples from that cancer type and tested on the remaining 30%. Univariate Kaplan–Meier analysis is performed on the filtered genes from the elastic net (with non-zero coefficients) to further diminish the number of genes used in the final model for calculating the combined immune score for the cancer type. The score is calculated by the sum of the expressions of the selected genes, weighted by their respective elastic net coefficients. (**b**) Thirteen immune-signature-associated genes that were identified from the elastic net, and univariate Kaplan–Meier analysis stratified TCGA GBM patients into two distinct clusters. (**c**,**d**) The immune score calculated from these 13 genes stratified TCGA GBM patients into better or worse prognostic groups for (**c**) DFS and (**d**) OS. *p*-value was calculated using Kaplan–Meier analysis. (**e**) In multivariate Cox regression analysis, immune score remains an independent predictor of DFS in GBM (TCGA cohort), along with age and MGMT methylation status. (**f**) The GBM immune score model using 13 prognostic genes significantly stratifies patients into better or worse prognostic groups in an independent validation cohort from the REMBRANDT project. *p*-value was calculated using Kaplan–Meier analysis.

**Figure 5 cancers-12-02476-f005:**
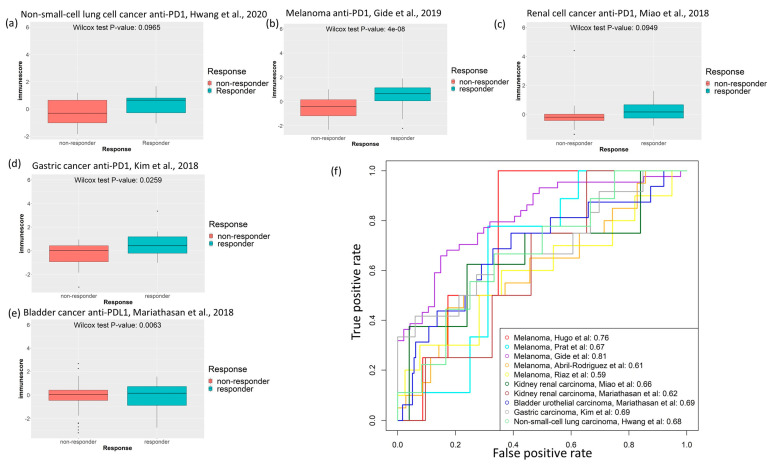
(**a**–**d**) Cancer-specific immune score is higher in patients with responsive tumors (CR/PR) than in patients with unresponsive tumors (SD/PD) in published data sets of anti-PD1-/anti-PDL1-treated tumors: (**a**) anti-PD1-treated non-small cell lung cancer [28], (**b**) anti-PD1-treated melanoma [22], (**c**) anti-PD1-treated renal cancer [26], (**d**) anti-PD1 treated gastric cancer [27], and (**e**) anti-PDL1-treated bladder tumors from the ImVigor210 trial [25]. (**f**) Immune score predicts objective response (RECIST) for anti-PD1/anti-PDL1 immunotherapy in a published data set of different tumor types; patients with CR or PR are considered as having responsive tumors, while patients with SD or PD are considered as having unresponsive tumors. Receiver operating characteristic curve for cancer-specific immune score predicting objective response is shown for: anti-PD1-treated melanomas [20,21,22,23,24], anti-PDL1-treated bladder tumors from the ImVigor210 trial [25], anti-PD1/PDL1-treated kidney tumors [25,26], anti-PD1 treated gastric cancer [27], and anti-PD1 treated non-small cell lung cancer [28].

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
