# Peer review of "Cancer-Specific Immune Prognostic Signature in Solid Tumors and Its Relation to Immune Checkpoint Therapies"

_cancers, 2020, doi:10.3390/cancers12092476_

Round 1
Reviewer 1 Report
In this comprehensive study the authors have identified cancer-specific prognostic immune score models with prognostic value for several cancer types. More important the authors demonstrate that these cancer-specific immune score models predict to some extend response to immune checkpoint inhibitor based immunotherapies. This was shown for anti-PD1-treated melanoma , anti-PD1- treated renal cancer and anti-PDL1-treated bladder tumors. There were strong trends but statistical significance was not reached.
Major points.
1.Immune checkpoint inhibition-based immunotherapies are most frequently applied in patients with NSCLC. Is there a specific prognostic immune score for NSCLC which could also predict clinical responses to anti-PDL1 or anti-PD1 treatments? If the authors could not identify one, this should be discussed.
- The traditional immunoscore is based mostly on frequencies of CD8+ and CD45RO+ T cells infiltrating the invasive margin and the tumor center of colorectal cancers. Such immune cell infiltration is closely associated with Th1 / IFNg signaling signatures. Moreover, pembrolizumab is now being used as immunotherapy in colorectal cancers with MSI. The authors should discuss why they don’t have data scRNAseq from colorectal cancers.
- As discussed above, the predictive value of the immune score models was shown in patients with 3 different cancer types receiving anti-PD1/PDL1 immunotherapies. There were trends for Wilcox test P values for all types of cancers. How many of the patients with CR or PR also developed immune resistance after treatments? Were these patients included in the group of those with CR or PR? This is important to know because it would give us an idea if these immune scores could also predict to a certain extend also adoptive immune resistance to immunotherapies.
Author Response
We thank the reviewer for the useful suggestions. Please see attached file for point by point explanation.

Reviewer 2 Report
The manuscript presents the results of RNA analyses from publicly available data. Several solid tumours were investigated. It was shown that 2 basic types of immune cell infiltration can be distinguished, one associated with NK, T and B cell functions, and one with monocytes, macrophages, dendritic cells and toll-like receptors. The data were examined with a sophisticated analysis tool and linked to the personal prognosis. The separate examination of the different tumor types was decisive. This revealed an association between immune scores and the response to checkpoint inhibitors. The methods are clearly described and comprehensible based on the publicly available data. The results are presented clearly, the graphs are labelled in a meaningful way. Current literature completes the paper.
Author Response
We thank the reviewer for taking time to evaluate our manuscript and very happy we are able to explain our findings to the satisfaction of the reviewer.
Reviewer 3 Report
Das et al. present cancer-specific signatures with value both in prognosis and in prediction of response to immunotherapies. In their paper, they present an interesting and novel concept related to the uniqueness of prognostic/predictive signature in individual (or group of) cancer types.
Multiple groups are still trying to find unifying signatures that can be applied across cancer types (the most successful being so far the T cell-inflamed GEP, by Ayers et al. 2017 and later applied in Cristescu et al. 2018 and Ott. et al. 2019). The results presented in this manuscript provide a novel view to this problem.
While the concept is novel and provide an answer to an important scientific question, the overall presentation of the manuscript (and results especially)can be improved significantly to improve the readability by a scientist who is not expert in bioinformatics or in cancer immunotherapy.
Some specific points are:
- In the results section, it is not explained well how results from chapter 2.1 are applied in chapter 2.2 (I understood it, but the authors do not explain clearly in the text). This can be improved with a paragraph at the end of 2.1 or at the beginning of 2.2.
- Explain more extensively in the results section some apparent paradoxes, e.g. a high immune cluster 1 appeared first to be associated with better prognosis in some cancer types (line 117-118), but then it seems like only the opposite is true (line 157-161). At first read, this may appear like a paradox - should be clarified in the text
- Many abbreviations in the results are not explained at their first appearance (e.g. page 2, line 60-62: all of these are non-standard abbreviations)
- The whole results 2.2 does not have a single Figure reference (it should be figure 2)
- Why only those 4 datasets of immune checkpoint-inhibitor treated patients were used? Multiple other datasets are available: Melanoma (Riaz, Gide, Abril-Rodriguez), Renal (McDermott), Gastric (Kim)
Author Response

(The authors gave the same response as above.)

Round 2
Reviewer 1 Report
none